# Coordinated inflammatory responses dictate Marburg virus control by reservoir bats

Jonathan C. Guito[1,6], Shannon G. M. Kirejczyk[1,2,4,6], Amy J. Schuh [1], Brian R. Amman [1], Tara K. Sealy[1], James Graziano[1], Jessica R. Spengler [1], Jessica R. Harmon [1], David M. Wozniak[3,5], Joseph B. Prescott [1,3] ✉ & Jonathan S. Towner [1] ✉

Bats are increasingly recognized as reservoirs of emerging zoonotic pathogens. Egyptian rousette bats (ERBs) are the known reservoir of Marburg virus (MARV), a filovirus that causes deadly Marburg virus disease (MVD) in humans. However, ERBs harbor MARV asymptomatically, likely due to a coadapted and specific host immunity-pathogen relationship. Recently, we measured transcriptional responses in MARV-infected ERB whole tissues, showing that these bats possess a disease tolerant strategy that limits pro-inflammatory gene induction, presumably averting MVD-linked immunopathology. However, the host resistant strategy by which ERBs actively limit MARV burden remains elusive, which we hypothesize requires localized inflammatory responses unresolvable at bulk-tissue scale. Here, we use dexamethasone to attenuate ERB pro-inflammatory responses and assess MARV replication, shedding and disease. We show that MARV-infected ERBs naturally mount coordinated pro-inflammatory responses at liver foci of infection, comprised of recruited mononuclear phagocytes and T cells, the latter of which proliferate with likely MARV-specificity. When pro-inflammatory responses are diminished, ERBs display heightened MARV replication, oral/rectal shedding and severe MVD-like liver pathology, demonstrating that ERBs balance immunoprotective tolerance with discreet MARV-resistant pro-inflammatory responses. These data further suggest that natural ERB immunomodulatory stressors like food scarcity and habitat disruption may potentiate viral shedding, transmission and therefore outbreak risk.

Growing appreciation for bats as asymptomatic reservoirs of human pathogens has fostered keen interest in understanding bat biology and ecology of these animals, including host immunity[1–3]. One of the best-characterized bat-zoonosis relationships is the Egyptian rousette bat (ERBs, *Rousettus aegyptiacus*), the only known reservoir host of

Marburg virus (MARV, family *Filoviridae*), which causes deadly Marburg virus disease (MVD) in humans and non-human primates (NHPs)[4–8]. Viable MARV has been repeatedly isolated from ERBs in sub-Saharan Africa, which has experienced numerous sporadic lethal MVD outbreaks since 1967[4,6,7], most recently in Equatorial Guinea and

[1]Viral Special Pathogens Branch, Division of High-Consequence Pathogens and Pathology, Centers for Disease Control and Prevention, Atlanta, GA 30329, USA. [2]Division of Pathology, Emory National Primate Research Center, Emory University, Atlanta, GA 30329, USA. [3]Center for Biological Threats and Special Pathogens, Robert Koch Institute, 13353 Berlin, Germany. [4]Present address: StageBio, Mount Jackson, VA 22842, USA. [5]Present address: Virology Department, Bernhard-Nocht-Institute for Tropical Medicine, 20359 Hamburg, Germany. [6]These authors contributed equally: Jonathan C. Guito, Shannon G. M. Kirejczyk. ✉e-mail: prescottj@rki.de; jit8@cdc.gov

Tanzania. Although MARV replicates in many tissues of infected ERBs and is shed at transmissible levels, only minimal microscopic tissue lesions are observed in these bats, with no clinical signs of inflammatory disease[9–12].

In MARV-infected primates, aberrant immune responses underlie MVD, resulting from direct viral protein antagonism of interferon (IFN) responses, disturbance of macrophage (MΦ) and dendritic cell (DC) functions (early target cells of MARV), skewing of normal innate and adaptive T cell responses, and induction of lymphocyte apoptosis[13,14]. Immune response dysregulation and extensive MARV replication in primates culminates in an uncontrolled, hyper-inflammatory state that disrupts vascular integrity[6,13]. Severe liver functional alterations and pathology are central to disease manifestations, contributing to development of often-fatal disseminated intravascular coagulopathies (DIC)[5,6,13].

Recent studies on bats have sought to understand how their immune systems and responses allow them to host pathogens unharmed, revealing distinct genomic and transcriptomic features indicative of a coevolutionary arms race with ancient viruses over millennia that is often specific to a particular bat species and its resident virus, including ERBs with MARV[2,7,15–18]. Some immunological properties of ERBs have been identified, including expansion of major histocompatibility (MHC) loci, diversified natural killer receptor loci and weaker induction of IFN genes[16,18,19]. ERBs inoculated with MARV mount virus-specific immune responses, demonstrated by anti-MARV antibody development and immunological memory to reinfection, although these antibodies are non-neutralizing[16,20–22]. Previously, we concluded that the lack of immunopathology seen in MARV-infected ERBs was explained by "disease tolerance," a protective infection defense strategy focused on controlling inflammatory response activation rather than on actively controlling virus burden, a strategy instead known as "host resistance"[9,23]. This was supported by our tissue-level transcriptional analysis that showed innate immune gene responses are induced upon MARV infection but consist mostly of canonical IFN signaling genes (ISGs) and very few significant pro-inflammatory genes[9], a striking difference from the severe cytokine-mediated pathology seen in humans and NHPs with MVD, and also from the high constitutive IFN expression observed in other bat hosts like *Pteropus alecto*, one of the species-specific immune features that may be corollary to adaptation to the rigors of flight[2,5,14,24].

Together, these efforts suggested how bats, including ERBs, broadly interact with their resident viruses, provided a framework for how illness is avoided, and offered evidence for a widely-held assumption that traditional inflammatory responses are less consequential in this unique class of mammals. However, in large part due to the paucity of bat-specific research tools (e.g., monoclonal antibodies, functional assays, tissue culture systems, etc.) that precludes detailed mechanistic approaches, the scope of such studies in bats has often been observational, low resolution and focused more on disease tolerance in defining bat defenses[1–3,16,25,26]. Indeed, for ERBs, fundamental questions remain, notably whether inflammatory responses, which remain poorly characterized, in fact play a host resistant role in actively controlling MARV replication, and more broadly if perturbations in immune status, possibly affected by ecological stressors like food scarcity, comorbidities, habitat disruption, etc., affect viral replication, disease and transmission. Here, we hypothesize that, despite minimal inflammatory gene regulation observed at a whole tissue scale[10,11,27], localized pro-inflammatory response induction is essential to limit MARV replication in ERBs, as found in other non-bat mammalian hosts[28–31]. We reasoned that experimentally driving the immune status of ERBs toward an overly immune tolerant, anti-inflammatory state in which pro-inflammatory responses are suppressed would abrogate this viral control, leading to altered cell tropism, enhanced replication and shedding, and potentially MVD-like

pathology[28,29,31,32]. Through a novel use of the glucocorticoid (GC) drug dexamethasone (Dex)[28,29,31] in bats and state-of-the-art molecular pathology tools to target key representative, highly-conserved immune markers[33–35], we show that discreet pro-inflammatory responses control MARV during ERB infection, and that these findings support the idea that immunomodulatory ecological stressor conditions could ostensibly undermine this control in wild bat populations[3].

## Results

### ERB cellular responses and pathology upon MARV infection

Liver and secondary lymphatic tissues (spleen and lymph nodes) are critical for immune response development and MVD onset in spillover hosts (e.g., humans)[5,6,13,14] and support the highest MARV replication in ERBs[9,11]. Previous histopathological inspection of ERB tissues was limited to hematoxylin/eosin (H&E) staining and immunohistochemical (IHC) detection of MARV antigen due to a lack of ERB-compatible reagents[10,11]. To investigate ERB inflammatory responses convincingly, we examined viral tropism along with multiple relevant immune cell populations via cross-reactive antibodies or species-specific RNA probes following experimental MARV inoculation, with molecular targets selected that are functionally integral to each immune cell type and highly evolutionarily conserved across mammals and/or vertebrates[33–35]. To evaluate MARV-mediated immunologic and pathologic changes, ERBs were euthanized 6 days post-inoculation (DPI) at peak MARV loads typically observed in these bats, or 12DPI when MARV is largely resolved, time points established by previous ERB infection studies[9–12].

H&E staining corroborated previous observations that characterized tissues in uninfected and MARV-infected bats[10,11]. An increased prevalence of discrete inflammatory foci of mononuclear cells were observed in MARV-infected bat livers at both 6 and 12DPI, often containing hemorrhages and low numbers of apoptotic or necrotic hepatocytes, while some, but not all, infected bats had remaining mild to moderate inflammatory cell infiltrates still present in skin at the inoculation site. No notable histopathologic lesions were found in any other tissue analyzed. To better resolve distribution of specific immune cell types in the ERB hepatic foci, we performed IHC on livers from uninfected and MARV-infected bats with species-cross-reactive antibodies targeting mononuclear phagocytes (MNPs, Iba1), T cells (CD3e) and B cells (CD79a) (Fig. 1a–h). Uninfected bat livers contained numerous elongated MNPs morphologically consistent with Kupffer cells (KCs), and occasional T cells within the sinusoids. Inflammatory foci, sparsely observed in MARV-infected bats, consisted primarily of MNPs (monocytes and KCs), with fewer T cells and rare B cells. Foci were absent in uninfected bats. MNPs were more abundant in infected ERBs than in uninfected bats at both time points. Although total abundance of T cells was comparable in uninfected bats and MARV-infected bats at 6DPI, there was a 12-fold increase in T cell signal at 12DPI, with cells distributed in scattered inflammatory foci and throughout the sinusoids (Fig. 1k). At both time points, staining for the cell proliferation marker Ki-67 associated predominantly with lymphocytes within foci that, based on morphology and the limited B cell abundance, are presumably T cells (Fig. 1i, j), while fewer Ki-67+ lymphocytes were also found in sinusoids at 12DPI, together suggesting that both liver-recruited and circulating T lymphocytes were proliferating in response to MARV antigen, and were most likely MARV-specific. To determine if MARV was associated with hepatic inflammatory foci, we designed novel in situ hybridization (ISH) probes detecting MARV *VP40/NP* and ERB-specific monocytes (*CD14*), T cells (*CD3e*) and B cells (*CD79a*) at 6DPI (Fig. 1l–n and Supplementary Table 1). Scattered MARV RNA was present in MNPs within foci, where MARV+ cells displayed MΦ morphology and were often *CD14*+ (Fig. 1l), as well as in cytoplasm of scattered hepatocytes often adjacent to foci, where MARV distribution was typically membranous.

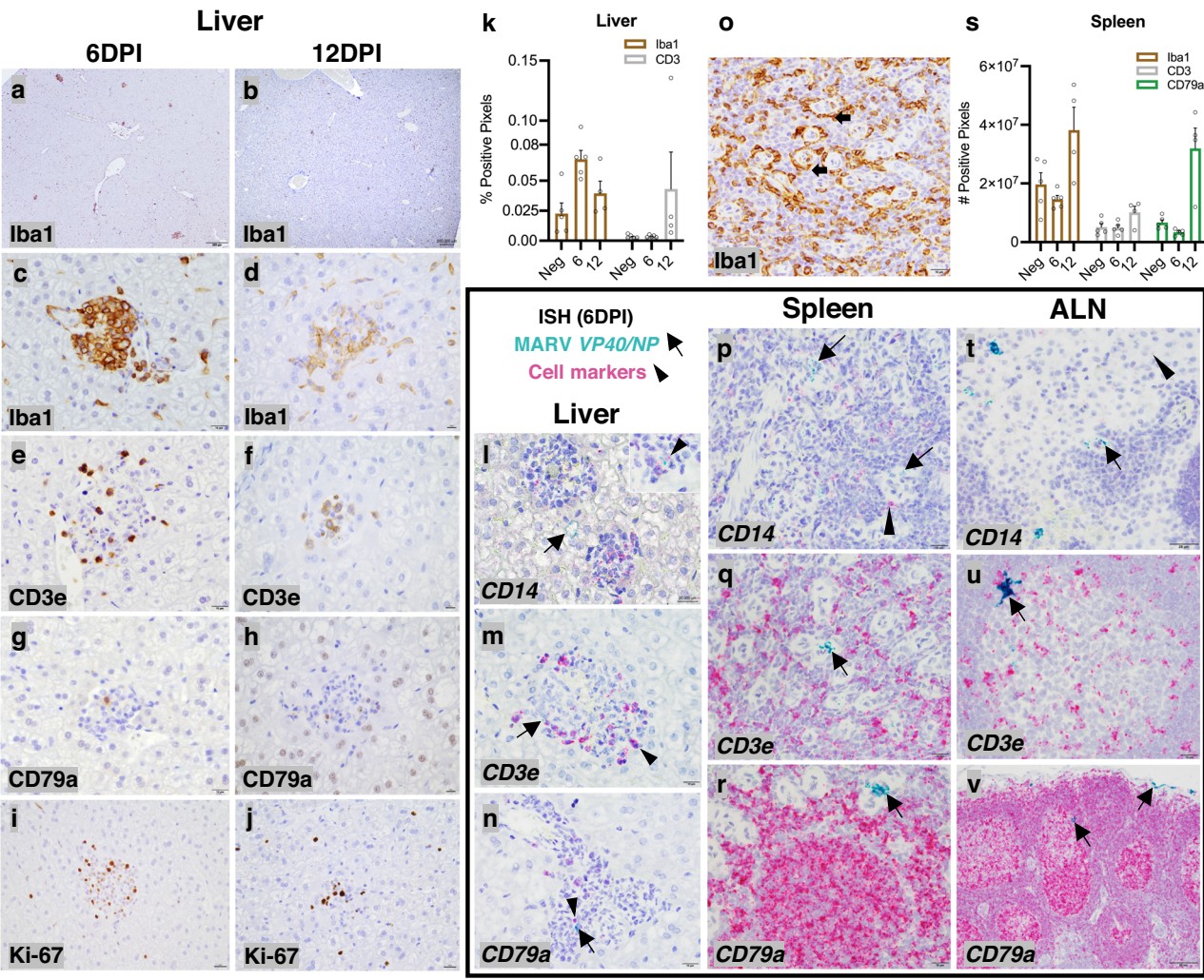

**Fig. 1 | Marburg virus (MARV) infects mononuclear phagocytes (MNPs) and hepatocytes in Egyptian rousette bats (ERBs) with MNPs and T cells recruited to hepatic MARV⁺ inflammatory foci. a–j** Immunohistochemistry (IHC) of hepatic immune cell populations in normal MARV-infected bats at 6 and 12 days post-infection (DPI, *n* = 5 bats/cohort and 4 bats/cohort, respectively). Iba1: Ionized calcium-binding adapter molecule 1, MNP marker; CD3e: T cell marker; CD79a: B cell marker; Ki-67: cell proliferation marker. Brown: 3,3'-Diaminobenzidine (DAB) chromogen. **a, b** Iba1. Scale bars: 200 μm. **c–h** Iba1, CD3e and CD79a. Scale bars: 10 μm. **i, j** Ki-67. Scale bars: 20 μm. **k** Whole-slide image analysis quantitation of hepatic MNPs and T cells. Bars: group means (*n* = 5) ± standard error of the mean (SEM). Data points: open circles represent individual bats. **l–n** Duplex in situ hybridization (ISH), probes targeting MARV *VP40/NP* (viral protein 40/nucleocapsid protein) in green (arrows) and immune cell markers (*CD14* [monocyte marker], *CD3e* and *CD79a*) in red (arrowheads). Scale bars: 20 μm (**l**) or 10 μm (**m, n**). **o** Iba1 IHC, spleen, 6DPI. Scale bar: 10 μm. **p–r** Duplex ISH, MARV and immune cell markers. Scale bars: 10 μm. **s** Whole-slide image analysis quantitation of splenic MNPs, T cells and B cells. Bars: group means (*n* = 5) ± SEM. Data points: open circles represent individual bats. **t–v** Duplex ISH, MARV and immune cell markers, axillary lymph node (ALN). Scale bars: 20 (**t**), 10 (**u**) and 50 μm (**v**). Hematoxylin counterstain (all IHC/ISH assays). Source data are provided as a Source Data file.

In red pulp of bat spleens, punctate MARV RNA signal was detected within the walls of sheathed capillaries (ellipsoids), likely in cytoplasm of slender MNP cell processes that interdigitate between endothelial cells (ECs), or in cytoplasm of ECs themselves (Fig. 1o–r). In axillary lymph nodes (ALNs) (Fig. 1t–v), MARV-infected cells were located within the sinuses, around small blood vessels, in the capsule and in follicles, and were morphologically consistent with MNPs. In follicles, infected cells resembled follicular DCs with numerous branching cytoplasmic processes (Fig. 1u, v). Lymphocyte (*CD3e* and *CD79a*) RNA staining aligned with the expected architecture of these lymphatic tissues and did not contain MARV; in the spleen, while quantities of T cells marginally increased between 6–12DPI, those of B cells substantially expanded, suggesting B cell maturation and proliferation (Fig. 1q–s, u, v). Unlike in bat livers, where limited focal inflammation was evident, there were no pathological tissue alterations in spleens or ALNs of infected ERBs.

## Dex immunomodulation of naïve ERBs

We next sought to better determine if inflammatory responses, such as those that appear to be elicited at hepatic foci, are required for controlling MARV infection in ERBs, and if suppression of these responses would lead to increased virus replication/shedding, altered cell tropism and ultimately disease. We therefore chemically modulated ERB immunity using the well-characterized GC drug Dex, known to block TNF and NF-κB responses, deplete lymphocytes and other immune-related cells, disrupt pro-inflammatory immunocellular activities and overall skew the immune system toward an immune tolerant, anti-inflammatory state[28,29,31]. We first assessed efficacy of Dex immunomodulation (IM) to deplete immune cells in naïve ERBs (Fig. 2a). We quantitated white blood cells (WBCs) at serial time points in uninfected IM bats compared to those in negative control (Neg) bats. WBC counts in IM bats were significantly reduced by more than 57% on day (d)4 post-IM, a level of depletion that was sustained until euthanasia. Dex

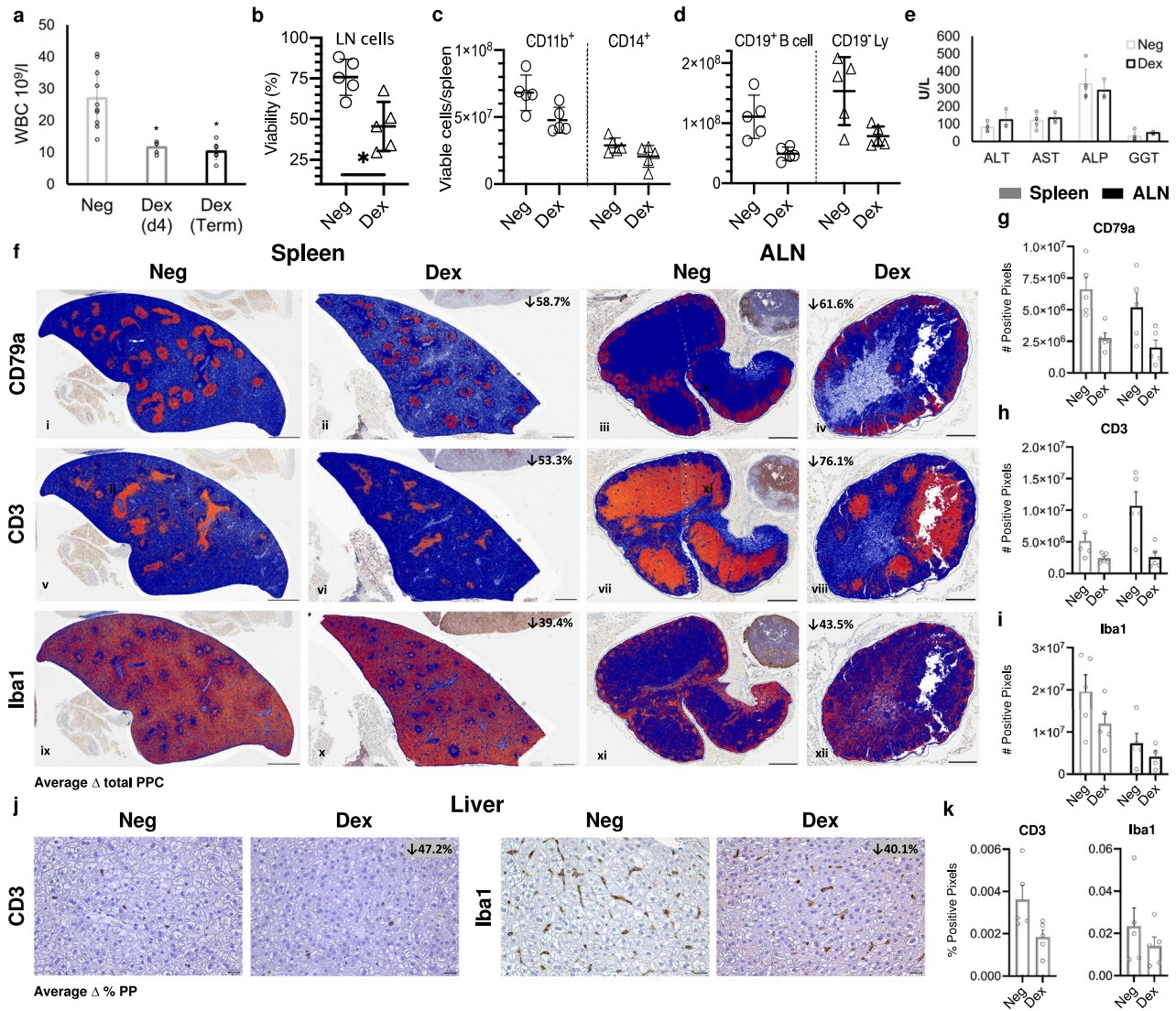

**Fig. 2 | Immunomodulation (IM) by dexamethasone (Dex) depletes immune cells in ERBs. a** White blood cell (WBC) counts, uninfected control (Neg) and uninfected IM (Dex) bats (day [d]4 and terminal d8 [Term], *n* = 5 bats/cohort; Neg counts averaged across both time points). Bars: 10⁹ cells per liter (l) of whole blood ± SEM. Data points: open circles represent individual bat samples. **b** Percent viability of ALN single-cell suspensions. **a**, **b** Two-tailed Student's *t* tests were applied for each grouping; *$p < 0.05$. **c**, **d** Flow cytometry of splenic single-cell suspensions. **c** Dendritic cell (DC, CD11b⁺) and monocyte (CD14⁺) populations. **d** B cell (CD19⁺) and T cell (CD19⁻ Ly [lymphocyte]) populations. **e** Liver-related blood chemistry in Neg bats (*n* = 5) and IM (Dex) bats (*n* = 3). ALT: alanine amino-transferase, AST: aspartate aminotransferase, ALP: alkaline phosphatase, GGT: gamma-glutamyl transferase. Bars: mean units (U) per liter (l) ± SEM. Data points: open circles represent individual bats. **f** IHC of Dex-mediated immune cell depletion. Digital markup images show B cell (CD79a⁺) (i–iv), T cell (CD3e⁺) (v–viii) and MNP (Iba1⁺) (ix–xii) pixels (orange) and negative pixels (blue). Scale bars: 500 μm. Brown: DAB chromogen with hematoxylin counterstain. Percent decreased immune cell staining in IM bats versus Neg bats shown in corners of ii, iv, vi, viii, x and xii. PPC positive pixel count. Whole-slide image analysis quantitation of CD79a (*n* = 5, **g**), CD3e (*n* = 5, **h**) and Iba1 (*n* = 5 except ALN of IM bats [*n* = 4], **i**). Bars: group means ± SEM. Data points: open circles represent individual bats. **j** IHC of hepatic immune cell populations in Neg and IM (Dex) bats. Left: CD3e. Right: Iba1. Scale bars: 20 μm. Percent decreased immune cell staining in IM bats versus Neg bats shown in corners of IM bat images. PP pixel positivity. **k** Whole-slide image analysis quantitation of hepatic CD3e and Iba1. Bars: group means (*n* = 5) ± SEM. Data points: open circles represent individual bats. See Supplementary Fig. 2 for flow gating strategy. Source data are provided as a Source Data file.

was well-tolerated by ERBs during the full time course, with no apparent changes in appetite or behaviors, no alterations in liver-related blood chemistry values (Fig. 2e), and no microscopic evidence of secondary bacterial infections.

Upon necropsy, spleens and ALNs of IM bats were 40–50% smaller than of controls. Cytometric analysis of ALN cells showed 40% lower cell viability in IM bats (Fig. 2b). DCs (CD11b⁺) were decreased by 35%, monocytes (CD14⁺) by 28% and B cells (CD19⁺) by more than half (Fig. 2c, d). We then gated on total CD19⁻ lymphocytes, expecting most were T cells with a small subset of other lymphocyte subtypes (e.g., natural killer T [NKT] cells), and these were also reduced by more than half. To inspect cell depletion microscopically, we performed IHC in

the spleen, ALN and liver using the antibodies targeting MNPs, T cells and/or B cells and quantified staining (Fig. 2f–k). IM bats showed 39.4–58.8% decreased splenic, 43.5–76.1% decreased ALN and 40.1–47.2% decreased hepatic immune cell staining compared to controls, congruent with the cytometric data, and correlating to the marked decreased cellularity seen in H&E-stained white pulp (spleens) and cortex (ALNs) (Fig. 2f).

## MARV viral loads are increased in IM bats
To temporally define if and when ERB inflammatory responses exert anti-MARV effects during infection, we initiated Dex immunomodula-tion simultaneously with MARV inoculation and euthanized cohorts of

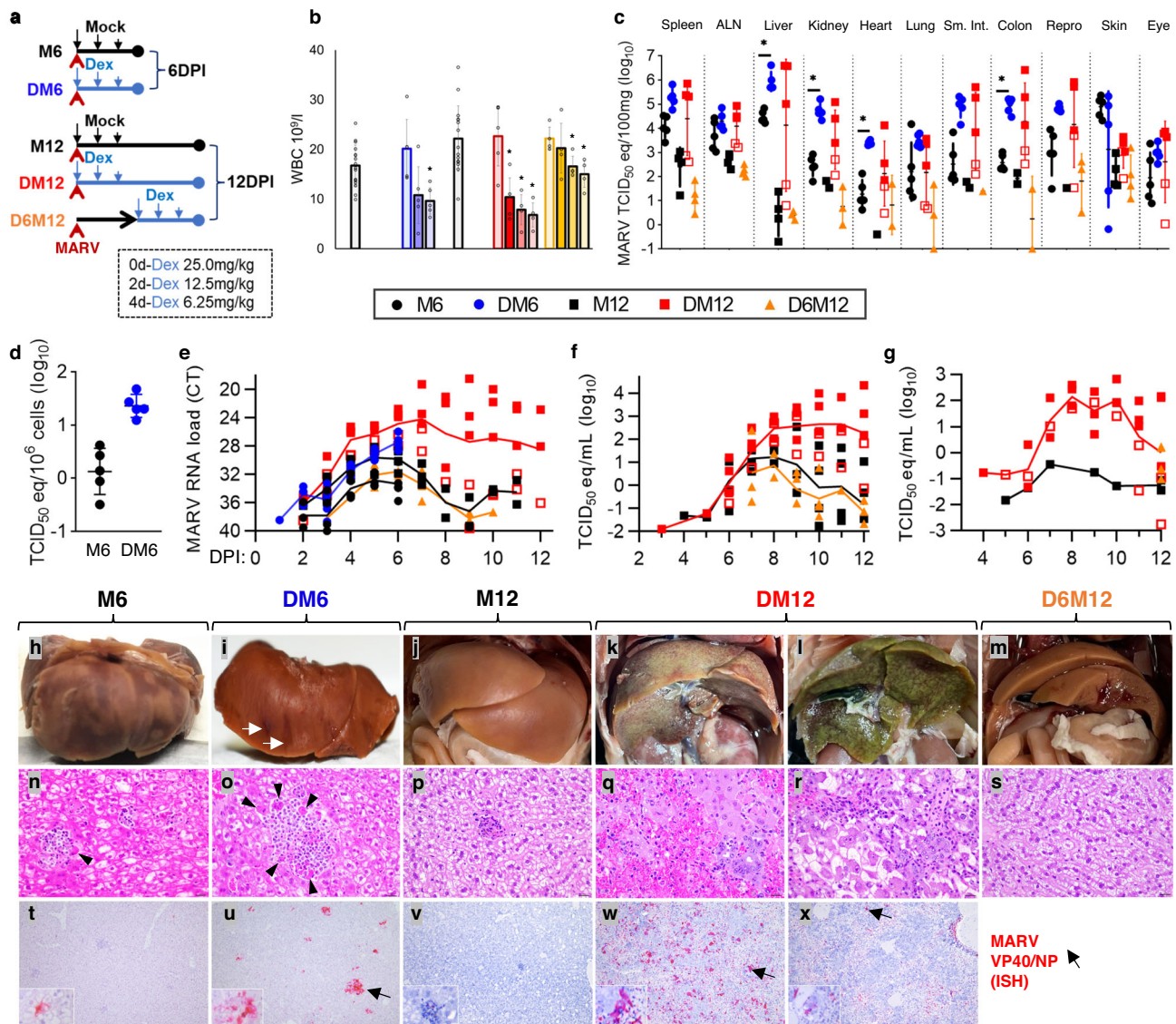

**Fig. 3 | IM bats exhibit exponentially higher MARV replication and severe MVD-like liver pathology. a** Experimental design. ERB cohorts: M6/M12 = MARV-inoculated, euthanized 6/12DPI; DM6/DM12 = simultaneous initial Dex treatment/MARV inoculation, euthanized 6/12DPI; D6M12 = MARV-inoculated, initial Dex treatment 6DPI, euthanized 12DPI. $n = 5$ bats/cohort except M12/D6M12 ($n = 4$). **b** WBC counts. Bars: mean counts ± SEM at 0/4/6DPI for DM6 bats ($n = 5$ bats/DPI; bar: 3-day mean for M6 bats [$n = 5$ bats/DPI]) and 0/6/9/12DPI for DM12 ($n = 5$ bats for 0/6DPI, $n = 4$ bats for 9/12DPI) and D6M12 bats ($n = 4$ bats/DPI; bar: 4-day mean for M12 bats [$n = 4$ bats/DPI except 12DPI ($n = 3$)]). Data points: open circles represent individual bat samples. Two-tailed Student's $t$ tests were applied for each time point compared to 0DPI within each cohort; *$p < 0.05$. Individual viral loads for: **c** Tissues ($n = 5$ bats/cohort/tissue except M12/D6M12 [$n = 4$]), **d** Splenic cells ($n = 5$ bats/cohort), **e** Blood (daily), **f, g** Oral (**f**) and rectal (**g**) shedding (daily). Data points: each sample indicated by cohort color/shape; open boxes: low-replicating

ERBs at 6DPI (DM6) or 12DPI (DM12). For another cohort, we delayed Dex IM until peak viremia on 6DPI and then euthanized bats at 12DPI (D6M12). Blood was collected daily until euthanasia to quantify the effect of IM on MARV replication, and viral loads in IM bat tissues were compared to those from time point-matched, MARV-infected control (mock-IM) bats (M6/M12 cohorts) (Fig. 3a). No infected IM bats showed overt clinical signs of Dex toxicity or viral disease (e.g., significant weight or temperature deviations) that would have merited premature euthanasia, although a few animals displayed modestly reduced appetence and, especially toward 12DPI, less apparent wound-healing

DM12 bats; lines: geometric mean trend except for (**e**) mean cycle threshold (Ct) trend; **c, d** vertical bars: geometric standard deviation (SD); **c** Two-tailed Student's $t$ tests were applied for each grouping as indicated; *$p < 0.05$. Sm. Int.: small intestine. **h–x** Gross, microscopic, ISH and ultrastructural liver pathology. **h–m** Gross appearance of formalin-fixed bat livers. M6 (**h**), M12 (**j**) and D6M12 (**m**) livers grossly normal. Hemorrhages in DM6 liver shown with white arrows (**i**). DM12 bats (**k**, **l**) show severe diffuse liver pathology. **n–s** Hematoxylin/eosin (H&E) staining. DM6 bats (**o**) show more hepatocellular apoptosis/necrosis (arrowheads) than M6 bats (**n**). M12 (**p**) and D6M12 (**s**) liver largely unremarkable with sparse inflammatory foci. High-replicating DM12 bats (**q**, **r**) show massive hepatocellular necrosis/hemorrhage (**q**) or parenchymal collapse (**r**), mononuclear cell inflammation and a ductular reaction. Scale bars: 20 μm. **t–x** MARV ISH, liver of indicated bat cohort. Scale bars: 100 μm (main), 10 μm (inset). Source data are provided as a Source Data file.

at wing venipuncture sites from which obtaining blood became more difficult. One bat, #1, was found on 12DPI to have recently died in its cage, but previously had not shown obvious signs of distress, illness or behavioral changes warranting euthanasia. As with uninfected IM bats (Fig. 2), DM6/DM12 bats showed significant and sustained WBC reductions through endpoint (-52–70%), while D6M12 bats showed more modest but still significant WBC depletion (up to -32%) (Fig. 3b).

Compared to M6/M12 mock-IM bats, MARV RNA loads were exponentially higher in multiple tissues and blood of IM-concurrent DM6/DM12 bats, but not in IM-delayed D6M12 bats, which instead

resembled M12 bats (Fig. 3c–e). Replication was often $10^1$–$10^2$ times higher in DM6/DM12 bats, most prominently in the liver, kidney and colon at both time points, heart and splenic single-cell suspensions at 6DPI, and spleen, ALN, small intestine, gonads and eyes at 12DPI (Fig. 3c, d). Indeed, mean hepatic MARV levels were $10^3$ times higher in DM12 versus M12 bats, with $10^5$ times more virus ($-4 \times 10^6$ TCID$_{50}$ equivalents/100 mg) in the cohort's two highest replicators (#1 and #2). In the blood, MARV RNA was higher in DM6/DM12 ERBs than in M6/M12 or D6M12 bats from 2DPI, nearly $10^2$ times higher by 6DPI, $10^3$ times higher by 10DPI (at which point virus in the other cohorts was virtually undetectable) and at $>10^2$ TCID$_{50}$ eq./ml (up to $-10^5$ TCID$_{50}$ eq./ml) by 12DPI, when the other cohorts were MARV-negative (Fig. 3e). While the effect of IM on MARV replication in DM6/DM12 bats was clear, we observed pronounced bat-to-bat variation in DM12 bats beyond 6DPI, with #1, #2 and #3 consistently supporting increased MARV loads (high-replicators, or high-rep) and #4 and #5 almost M12-like (low-replicators, or low-rep).

Shedding of MARV RNA from the oral and rectal mucosa of DM12 ERBs was also exponentially higher and more prolonged compared to M12 and D6M12 bats (Fig. 3f, g). Oral/rectal shedding was absent or barely detectable until 7DPI, quickly increased to $-10^2$–$10^{2.5}$ TCID$_{50}$ eq./ml by 8DPI and was roughly maintained through 10DPI rectally and 12DPI orally. This contrasts with oral shedding from M12 controls, which showed $10^1$ times lower virus than DM12 bats at their peak (8DPI), waning levels by 9DPI and near undetectability by 10DPI, and more evidently for rectal shedding, which was almost completely undetectable save one bat (#6) with minimal MARV RNA levels. Variation in DM12 MARV shedding was consistent with blood and tissue data, with #1 being the highest shedder (at $>10^4$ TCID$_{50}$ eq./ml orally, higher than inoculum) and #5 being the lowest.

## IM bat pathology reveals features resembling MVD

We next investigated how MARV-mediated pathology in liver and other tissues was altered in IM bats. Using the MARV-specific ISH probes to assess hepatic replication, viral RNA greatly and progressively increased in histological sections of DM6 and high-rep DM12 bats, spreading from numerous discrete inflammatory foci to nearly the entire hepatic parenchyma, compared to rare findings of MARV RNA and associated foci in M6/M12 bats (Fig. 3t–x). Livers of M6/M12, D6M12 and low-rep DM12 bats appeared grossly normal, while DM6 bat liver showed multifocal hemorrhages and high-rep DM12 bats (particularly #1 and #2) advanced further to severe, massive hepatic necrosis, hemorrhage and/or parenchymal loss in enlarged tan/dark-red liver (Fig. 3h–m). Gross hemorrhagic lesions observed for DM6 bat liver correlated microscopically with the extensive increase in hemorrhagic, MNP-containing inflammatory foci, with more abundance of apoptotic or necrotic hepatocytes (cytopathic effects, CPE), compared to the sparse foci seen for M6 bats (Fig. 3n, o). Livers of M12, D6M12 and low-rep DM12 bats similarly exhibited scarce inflammatory foci, whereas high-rep DM12 bats had massive hepatocellular necrosis with hemorrhage, lesions typically associated with MVD in spillover hosts (Fig. 3p–s). Death for DM12 bat #1 at 12DPI was caused by sepsis, likely occurring secondary to severe MARV-associated liver and/or colonic pathology, while bat #3, which showed moderate levels of MARV RNA mainly in areas of hepatic necrosis, had a shrunken, green liver with marked hepatic parenchymal loss, cholestasis and a prominent ductular reaction, indicative of severe MARV-induced liver injury and commencement of a regenerative response (Fig. 3l, r, x).

Assessing viral tropism across tissues, MARV RNA in M6 bats (at peak viremia) was limited histologically to the liver, spleen, ALN and skin at the inoculation site, with exceedingly rare viral RNA detected in residual hepatic foci and splenic white pulp by 12DPI in M12 bats, agreeing with prior observations[10,11]. However, in DM6/DM12 bats, virus was histologically resolvable in tissues typically found MARV+ but only via PCR: stomach (interstitium, epithelium), intestines (interstitium), colon (interstitium, epithelium), kidney (glomerulus, renal tubules), urinary bladder (smooth muscle, submucosa), adrenal cortex, heart (endocardium), salivary gland, lung, trachea, thymus, eye (conjunctiva), epidermis (inoculation site, others), ovary (granulosa) and uterus (smooth muscle), and expanding in DM12 bats to further include tongue and penile epithelium (notable examples shown in Supplementary Fig. 1). In the exocrine pancreas, adrenal cortex, kidney, colon, tongue, ovary and penis of high-rep DM12 bats, MARV RNA was associated with CPE (demonstrated in Supplementary Fig. 1f, j, m, n, z, aa). Full MARV ISH results from individual bats and tissues can be found in Supplementary Table 2.

## MARV cell tropism and immune cell responses in IM bats

Liver is naturally immune tolerant and shapes immunity in important ways; it is distinct from secondary lymphoid tissues and central to viral pathogenesis[5,6,13,14,36,37]. Given the profound hepatic MARV replication, associated CPE and severe MVD-like pathology observed in ERBs under immunomodulatory conditions, we further examined MARV cellular tropism and immune cell recruitment in the liver. In DM6 bats, the number of hepatic inflammatory foci per mm$^2$ increased fourfold compared to M6 bats, with foci in both cohorts comprised mainly of Iba1$^+$ MNPs (Fig. 4a, b). MARV ISH signal in DM6 bats increased in hepatocytes and MNPs within inflammatory foci and was additionally found in KCs (Fig. 4c). In high-rep DM12 bats, marked numbers of MNPs infiltrated areas of hepatic necrosis (Fig. 4d, i). As with M6 bats, DM6 foci contained fewer *CD3*$^+$ T cells compared to MNPs, but in high-rep DM12 bats, T cell numbers increased tenfold compared to M12 bats (Fig. 4e, f, i). *CD79a*$^+$ B cells were rare in livers of all bat cohorts, except in high-rep DM12 bats where mild numbers were observed in necrotic areas; neither T nor B cells colocalized with MARV RNA. Ki-67 immunostaining in DM6 bats was limited to T cells within foci, again indicative of proliferation in response to MARV infection, likely antigen-specific, while in DM12 bats, Ki-67 staining was extensive in T cells, proliferating ducts and hepatocytes (Fig. 4g, h), consistent with a regenerative response to tissue damage along with T cell responses.

Secondary lymphoid tissues are crucial for immune response development and are early sites of MARV replication in ERBs and primates[5,6,13,14]. Additionally, the splenic microvasculature is involved in the clearance of some circulating viruses[37,38]. Expectedly, DM6 bats had grossly smaller spleens than M6 bats, with histologic reductions in MNP, T and B cells (Fig. 4j). Further, DM6 bats showed no overt immune cell recruitment to spleen and MARV ISH probes did not colocalize with T or B cells. Like the tissue viral load data, MARV signal increased in DM6/DM12 bat spleens, but as in M6 bats, the majority of MARV RNA was contained within and immediately surrounding red pulp ellipsoids, mostly in Iba1$^+$ MNPs that were occasionally *CD14*$^+$ (Fig. 4k, l, o, p). In high-rep DM12 bat spleens, while general immune cell marker abundance was quantitatively similar to M12 bats (Fig. 4j), cellular distribution differed. Here, splenic lymphoid cell depletion persisted (as in DM6 and uninfected IM bats) (Fig. 4k), but a novel MARV-associated vasculopathy was observed, linked to vascular leakage and leukocyte accumulation, with severe red pulp congestion and/or hemorrhage (Fig. 4m, n). This vasculopathy was characterized by an increased number of rounded MNPs in thickened ellipsoid walls (Fig. 4n, q, r), suggestive of monocyte recruitment and vasculitis, possibly due to vascular antigen-antibody complex deposition.

## Cytokine responses in normal and IM bat infection

Given the cytokine dysregulation characteristic of MVD in filovirus spillover hosts such as humans[6,13,14,39] and the known immunomodulatory effects by GCs on cytokine signaling pathways[28,29,31], we sought to correlate spatial expression of classical representative pro-inflammatory (tumor necrosis factor, *TNF* and interleukin-6, *IL6*) and anti-inflammatory (*IL10*) cytokine markers in ERB livers and spleens to suppression of pro-inflammatory responses

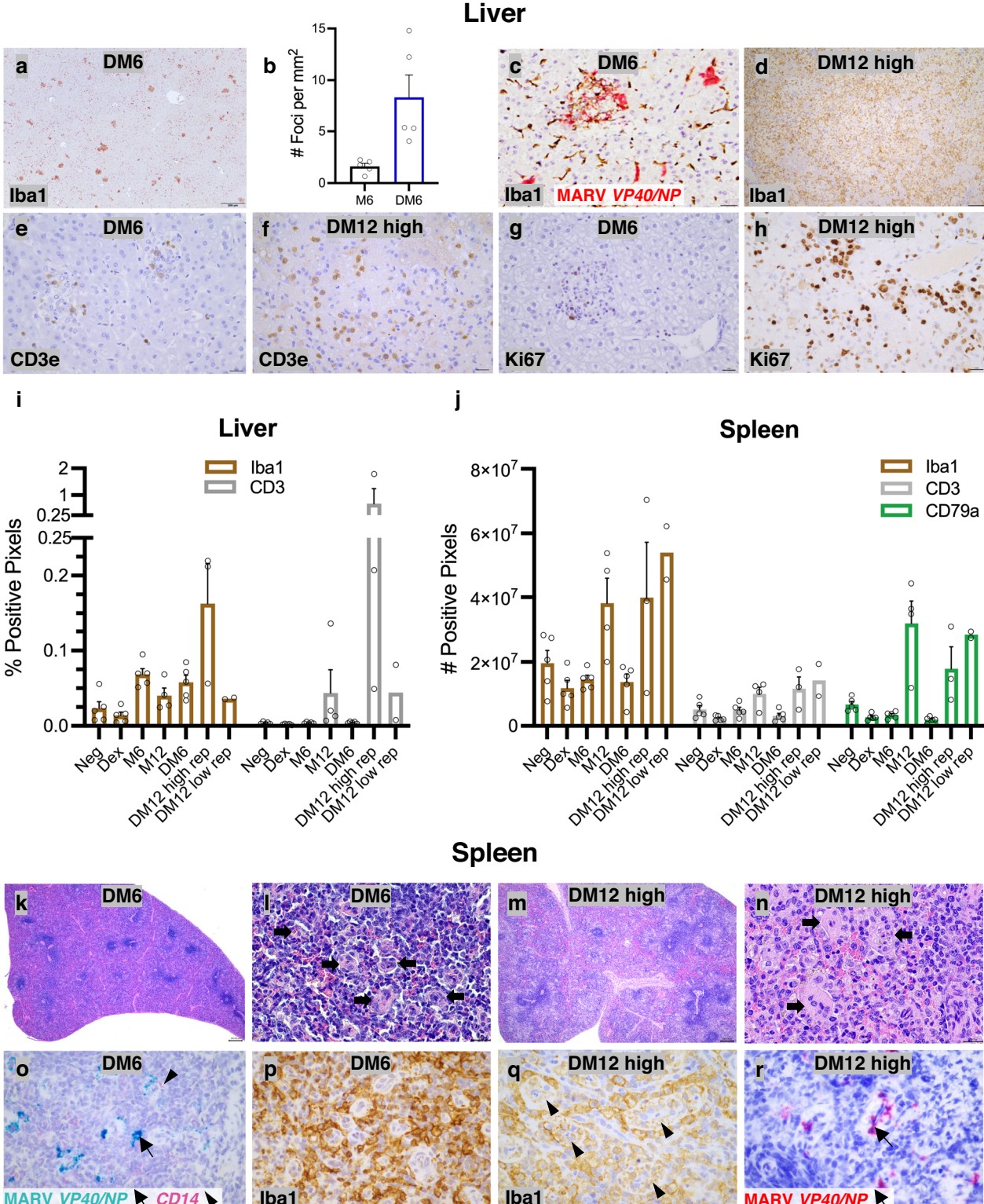

**Fig. 4 | MARV cell tropism and immunocellular responses in IM bats.** IHC/ISH of immune cell populations and MARV replication for indicated ERB cohorts. Brown: DAB chromogen (IHC) with hematoxylin counterstain (IHC/ISH). **a** Iba1 IHC. Scale bar: 200 μm. **b** Whole-slide image morphometric analysis of Iba1+ foci. Bars: group means ($n = 5$) ± SEM. **c** Dual MARV ISH (red) and Iba1 IHC (brown) demonstrates multifocal colocalization of MARV with sinusoidal Iba1+ Kupffer cells (KCs)/MNPs within an inflammatory focus. Scale bar: 10 μm. **d** Iba1 IHC. Scale bar: 50 μm. DM12 high: high-replicating bat. **e, f** CD3e IHC. **g, h** Ki-67 IHC. Scale bars: 20 μm (**e–h**). **i** Whole-slide image analysis quantitation of hepatic MNPs (Iba1) and T cells (CD3e). **j** Whole-slide image analysis quantitation of splenic MNPs (Iba1), T cells (CD3e) and B cells (CD79a). Bars (**i, j**): group means ($n = 5$ bats, except M12 [$n = 4$], high-

replicating [high-rep] DM12 [$n = 3$] and low-replicating [low-rep] DM12 [$n = 2$] bats) ± SEM (except for low-rep DM12 bats). Data points: open circles represent individual bats. **k–n** H&E staining (DM6 bats [$n = 5$], high-rep DM12 bats [$n = 3$]). **k** Dex-induced splenic lymphoid cell depletion. **l** Ellipsoids are present throughout the splenic red pulp (arrows); $n = 5$ bats. **m** Dex-induced splenic lymphoid cell depletion and red pulp congestion/hemorrhage; $n = 3$ bats. **n** Splenic ellipsoid walls are thickened and contain mononuclear cells (arrows); $n = 3$ bats. Scale bars: 80 μm (**k, m**) or 20 μm (**l, n**). **o** Duplex ISH, MARV in green (arrow) and *CD14* (monocyte marker; arrowhead) in red; $n = 1$ bat. **p, q** Iba1 IHC (DM6 bats [$n = 5$], high-rep DM12 bats [$n = 3$]). **r** MARV ISH; $n = 3$ bats. Scale bars: 10 μm (**o–r**). Source data are provided as a Source Data file.

by Dex. Indeed, these cytokines have been shown to be highly involved in dictating both inflammatory signaling and MVD onset/severity, as well as to be functionally conserved across the animal kingdom, with origins dating as far back as the Precambrian era[6,33–35,39]. In uninfected IM (Dex) bats, there was a general reduction in RNA for all three cytokines compared to naïve control (Neg) bats, with *TNF* in particular showing reduced expression, the exception being a modest increase in *IL10* in the liver (Fig. 5a, b). In M6/M12 spleens, *TNF* increased between 6 to 12DPI and *IL10* was increased at both time points compared to Neg bat expression, while in M6/M12 livers, both cytokines were increased at both time points, all consistent with an active and evolving immune response to normal MARV infection (Fig. 5a, b). In M6/M12 spleens, *TNF* and *IL10* were found primarily within the lymphoid compartment and in red pulp MΦ occasionally adjacent to MARV⁺ cells, respectively (Fig. 5c, d, g, h). Meanwhile, in high-rep DM12 bats, the splenic increase of *TNF* from 6 to 12DPI seen between M6 and M12 bats did not occur compared to DM6 bats and stayed at Neg-like levels, while in low-rep DM12 bats, *TNF* showed a more M12-like increase (Fig. 5a, c–f). *IL10* was higher in splenic red pulp MΦ of high-rep DM12 bats compared to M12 spleens but showed overall similar expression between MARV-infected cohorts (Fig. 5a, g–j). In DM6 livers, *TNF* and *IL10* were mildly increased in MΦ within inflammatory foci but again showed overall similar expression to M6 livers, despite the marked increase in viral RNA in often-adjacent MARV⁺ hepatocytes (Fig. 5b, k, m, o, q). Conversely, in high-rep DM12 bats, hepatic *IL10* expression was by far the highest of any infected bats and found in MΦ in areas of necrosis, whereas hepatic *TNF* in these same bats was only mildly increased, despite *TNF* signal found in multiple cell types including MΦ and regenerating duct epithelial cells (Fig. 5b, k–r). *IL6* was expressed at consistently low levels in livers and spleens of most bats, except for induction in the necrotic areas of high-rep DM12 livers, suggestive of tissue damage-induced local stimuli, and in the spleen of only the highest-replicating DM6 bat (#7), suggestive of bat-specific, virus-induced local stimuli (Fig. 5a, b).

## Discussion

Bats display an array of fates upon viral infection, ranging from severe disease or death (some lyssaviruses, possibly Lloviu filovirus) to non-productive viral clearance[2,15]. Viruses can also coevolve within specific bats, resulting in a virus-reservoir relationship, such as MARV with ERBs, whereby adaptations by both pathogen and host allow for asymptomatic replication, shedding, transmission and maintenance within the reservoir population[8,17,30]. MARV, along with closely-related but genetically distinct Ravn virus (RAVV), are the only known filoviruses capable of productive infection in ERBs, and the only human-pathogenic filoviruses for which a reservoir has been definitively identified[7,8,10,12]. We previously showed that ERBs mount whole tissue-level transcriptional responses to MARV consisting primarily of moderate canonical ISG induction but little significant pro-inflammatory gene upregulation[9], which contrasts with the runaway pro-inflammatory cytokine responses associated with human/NHP MVD[5,6,14,39].

The results presented herein redefine our understanding of MARV infection and host immunity in its bat reservoir. They demonstrate that a host resistant strategy of active and localized pro-inflammatory responses are necessary to control MARV infection, and that ERB defenses based on disease tolerance mechanisms alone[1,9,25,26] are insufficient for avoiding MARV pathogenesis and reducing viral burden. This MARV control appears to be largely hepatocentric, consisting of inflammatory hepatocellular MARV foci that recruit MNPs and T cells, the latter of which likely then proliferate in response to MARV infection. As hypothesized, attenuation of pro-inflammatory responses by Dex causes loss of immunologic control of MARV replication, leading to increased viral loads in tissues. Dex-mediated diminishment of basal and MARV-induced expression of pro-inflammatory *TNF*

(presumably with concomitant dysregulated signaling and immunocellular functioning), and reduced support from infiltrating immune cells depleted from circulation, liver and secondary lymphoid tissues, all likely contribute toward increased hepatocyte infection, foci generation, heightened virus shedding and potentially fatal tissue damage reminiscent of MVD in primates (e.g., massive hepatic necrosis and hemorrhage). Our work provides in vivo evidence that a naïve bat reservoir immunologically resists a virus it naturally harbors in a delicate balance, and that suppression of normal inflammatory and cellular responses ablates protection from viral disease and exacerbates shedding, and therefore potential for transmission and spillover.

During normal MARV infection of ERB livers, representative pro- and anti-inflammatory cytokines were expressed in MNPs adjacent to MARV⁺ cells, but not in infected cells themselves, perhaps due to paracrine signals emitted by infected cells, possibly in an IFN-independent manner, in response to attempts by MARV VP35 and VP40 to antagonize ISG induction[9,23,40]. Paracrine signaling by infected cells could also have been from another pro-inflammatory cytokine not measured in this study that subsequently induced *TNF* and/or *IL10* in a subset of surrounding cells, or alternatively, MNPs had been initially infected and then cleared MARV, but were still expressing low levels of cytokines at the time bats were euthanized. Given the rarity of viral RNA within normal M6/M12 inflammatory foci, compared to the exponentially higher amounts seen in immunomodulated DM6/DM12 bats (particularly in susceptible hepatocytes that likely also have responses antagonized by MARV), it is striking how these M6/M12 foci showed marked recruitment of MNPs and T cells, and stimulation of proliferative, likely T cell responses, yet only modest cytokine induction. This superb control by ERBs to prohibit MARV from hijacking responses that can lead to inflammatory disease, even with normal pro-inflammatory signaling unconstrained by the suppressive effects of Dex, underscores the precise relationship of ERB immunity with MARV infection, and the specialized role disease tolerance does play in protection from serious immunopathology[1,9,16,23,25,26]. On the other hand, the effects seen histologically in infected IM bats stress the importance of immunological balance, as a system pushed too far toward an immune tolerant and unresponsive state dampens the essential early inflammatory signaling it has been adapted to induce and regulate, including proliferation and activities of critical innate and adaptive immune cells. This can equally lead to serious virus-mediated pathology, even within its natural bat host, that still shows primate disease-like manifestations despite being devoid of its distinctive hyper-inflammatory response induction[28–31].

Unlike for DM6/DM12 cohorts, Dex had no effect in D6M12 bats (initial Dex treatment at 6DPI), which largely appeared indistinguishable from normal M12 bats. This demonstrates that MARV control by ERBs depends on the early activation of inflammatory responses, including likely adaptive responses, established within the first few days of infection. This is supported in DM6/DM12 bats by the recruitment and proliferation at hepatic foci of what are likely MARV-specific *CD3*⁺ T cells (as proliferation, ascertained by immunostaining Ki-67⁺, requires antigen presentation and secondary signals) and by the splenic expansion of B cells suggestive of maturation and proliferation, processes already initiated by 6DPI and thus mostly resilient to immunomodulation. Previous ERB studies further show that, despite a lack of antibody neutralization, MARV-specific antibodies are still produced and ERBs are nonetheless protected from rechallenge[20–22]. These earlier findings combined with our present histology data strongly suggest that, beyond their innate roles during acute infection, T cells and B cells also elicit potent adaptive memory responses to prevent MARV reinfection. Finally, the necessity of early response activation is evidenced by the exponential shift in MARV blood replication kinetics in DM6/DM12 bats observable at just 2DPI, indicating that normal resistance by innate cellular inflammatory responses within circulation or even upstream at the skin inoculation site has

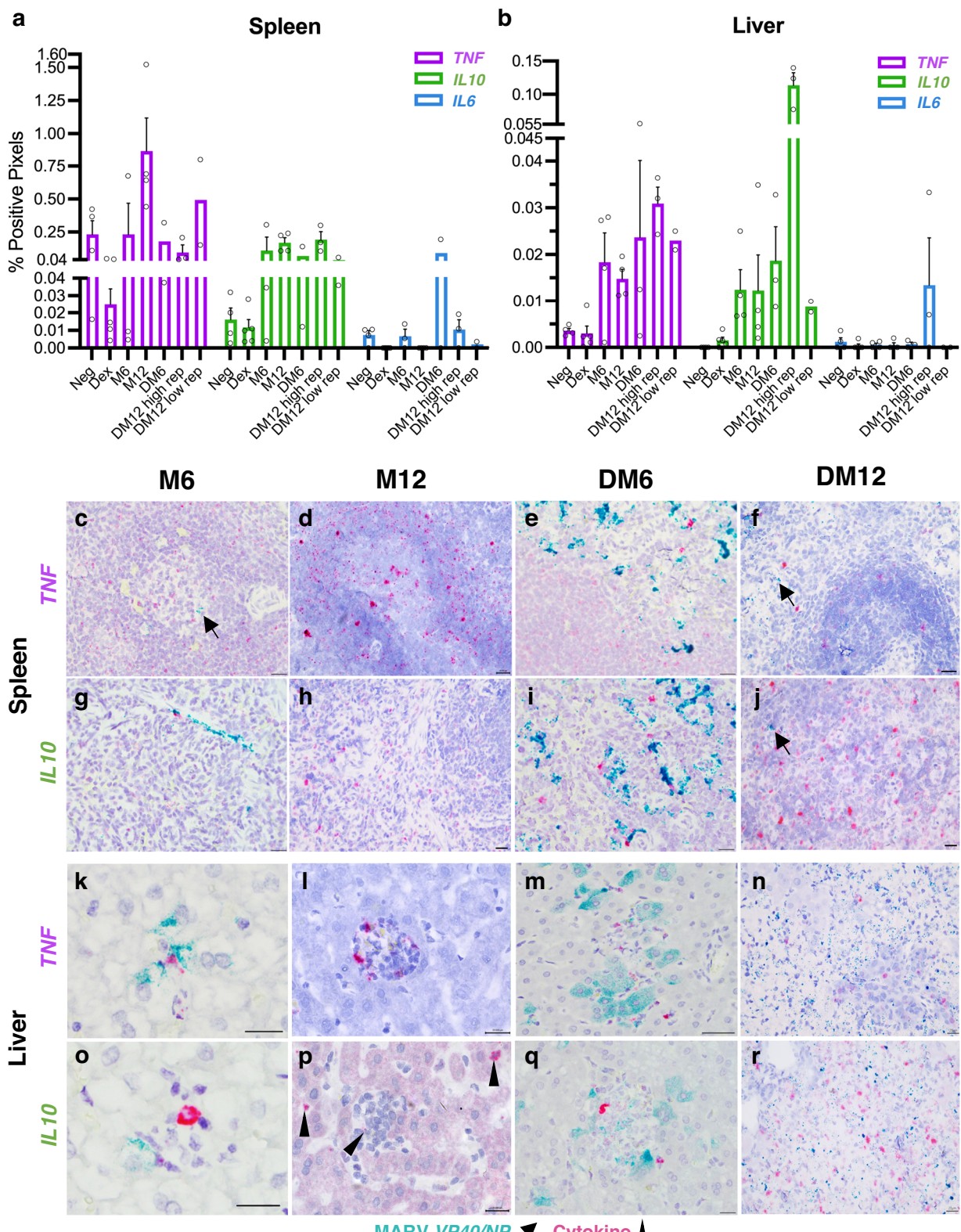

**Fig. 5 | Hepatic and splenic cytokine responses to MARV infection in normal and IM bats. a**, **b** ISH of pro-inflammatory (*TNF*, *IL6*) and anti-inflammatory (*IL10*) cytokine marker expression for indicated ERB cohorts, quantified by whole-slide image analysis. Bars: group means (Neg [*n* = 4], Dex [*n* = 5], M6 [*n* = 3 for liver and *n* = 4 for spleen], M12 [*n* = 4], DM6 [*n* = 3 for liver and *n* = 2 for spleen] and DM12 bats [*n* = 3 for high-replicating (high-rep) bats and *n* = 2 for low-replicating (low-rep)

bats]) ± SEM (except for DM6 spleen and low-rep DM12 bats). Data points: open circles represent individual bats. **c**–**r** Duplex ISH, MARV (green; arrows) and cytokine expression (red; arrowheads). Scale bars: 20 μm (**c**–**j**, **l**, **n**, **p**, **r**) or 10 μm (**k**, **m**, **o**, **q**). Hematoxylin counterstain. Source data are provided as a Source Data file.

already been chemically attenuated by Dex. This in turn presumably allows more virus to disseminate to tissues such as the liver that, due to Dex-induced cellular dysfunction and increased hepatocyte infection, can no longer effectively limit MARV replication.

The abundant MARV loads in blood and tissues and concomitant MARV-mediated disease seen in immunomodulated DM6/DM12 bats were facilitated through the multifactorial activities of Dex, an anti-inflammatory GC often used in transplant biology and treatment of severe SARS-CoV-2-mediated lung inflammation[41,42]. Dex has also been used in other immunosuppressive animal infection studies, including for mice infected by Zika virus and Syrian hamsters infected by Sin Nombre hantavirus[28,29]. As we saw for ERBs, Dex was well-tolerated in both rodent models, with minimal toxicity or clinical or histological changes, while showing clear depletion of WBCs and lymphocytes in the hamsters, and increased viral loads in the absence of inflammatory cell infiltrates in mouse tissue, suggestive of the pro-inflammatory response downregulation and shift toward an anti-inflammatory state characteristic of Dex[28,29]. TNF, a regulator of the NF-κB pathway and critical cytokine marker of severe filovirus disease in primates due to its uncontrolled activation[5,6,13,14,39], normally regulates pro-inflammatory responses in part by promoting such immune cell infiltration via activation of ECs and MΦ phagocytosis[39]. Dex, often through regulation via the GC receptor, inhibits pro-inflammatory TNF and IL6 production and NF-κB signaling, drives M1-to-M2 MΦ and Th2 polarization, augments the cytokine environment, stimulates autophagy, and promotes anti-inflammatory gene expression, as well as MNP (MΦ, DC, KC) and regulatory T cell (Treg) responses via modification of monocyte activation/maturation[28,29,31,43–45]. Evidence of M2/Th2 polarization in DM6/DM12 bats was apparent in uninfected mononuclear cells within liver foci, which showed more intense *IL10* expression than these same cells in M6/M12 bats. Further, stimulation of autophagy could underlie the increased hepatocyte infection, as it has been shown to enhance cell susceptibility to filoviruses[46]. Finally, IM bats showed *TNF* suppression in situ before and after MARV infection, suggesting that disrupted TNF-induced inflammation could no longer contribute to virus control. Indeed, the bat-to-bat variation seen among DM12 bats supports this, as low-rep bats with near-normal pathology show greater splenic *TNF* responses than high-rep bats, which instead show drastically increased hepatic *IL10* and likely tissue damage-induced *IL6*, a cytokine profile resembling that observed in humans with severe filoviral disease, emphasizing how IM-mediated suppression of early pro-inflammatory signaling can drive a normally-asymptomatic bat reservoir toward a more spillover host-like disease phenotype[47]. This bat variation is itself most likely explained by individual-specific immune recovery due to differing rates of Dex metabolism in the outbred, captive-bred ERBs and waning activity in tissues following final treatment at 4DPI. However, this IM bat variance strengthens the relationship in ERBs between immunocellular responses, MARV replication and pathogenicity, where DM12 bat #1 (the deceased bat) and #2 were the most immunodepleted with the highest viral loads and most extensive tissue damage. These observations highlight that ERBs, when deprived of their ability to mount appropriate inflammatory responses, are vulnerable to MVD-like pathology even with MARV loads several orders of magnitude lower than that found in sick NHPs preceding mortality[13], further indicating that disease tolerance has a more limited influence on overall viral pathogenesis in ERBs (and potentially, in other bat hosts) than previously appreciated.

While our study offers new insight into the relationship between pro-inflammatory responses and protection from virus-mediated disease in the context of a filovirus bat reservoir, it is not without caveats and limitations that must be considered. First, the mechanisms of Dex activity, while well-known, are not specific to any one cellular or molecular target, given suppression of IL6-, TNF- and NF-κB-mediated pro-inflammatory signaling affecting multiple downstream pathways

and immunocellular functions and depletion of multiple immune cell types. Our immunomodulatory analysis is thus meant to demonstrate that pro-inflammatory responses are, in fact, an essential antiviral component of normal MARV infection control in ERBs. However, any discussion at this stage about specific immune mechanisms directly responsible for inhibiting MARV replication remains speculative. Second, the extent of our analysis is constrained by a lack of resources including ERB-specific reagents and availability of juvenile bats, necessitating careful selection of representative time points, tissues and immune targets known to be critical to both immune control and filovirus disease progression. And third, although the annotated cellular and molecular targets we selected for our study (e.g., CD14) are functionally conserved across many mammalian, vertebrate and even invertebrate species (e.g., evidence of active TNF and related super-family members identified in coral, jellyfish and mollusks)[33–35], with any differences in ERBs likely ascribed to their transcriptional regulation or nuanced activities rather than core response role, nascent understanding of bat immunology means caution should nevertheless be taken when drawing conclusions about putative functional similarities between ERB targets and their counterparts in longer-studied mammalian animal models.

Finally, we propose that effects on MARV replication and shedding observed here in IM bats may similarly occur in wild ERBs experiencing immunomodulatory stress (e.g., possibly due to comorbidities, effects of climate change, etc.)[3]. Our study has broad implications for disease ecology and public health by showing that conditions that compromise a natural bat host's immune system can directly influence virus replication, shedding and presumably spillover. DM12 bats showed greatly increased rectal and oral MARV shedding starting from 7–8DPI, both sustained at high levels through study endpoint, with much more rectal shedding than is typical during infection in otherwise healthy ERBs[10,12,48,49]. This oral/rectal super-shedding seen in DM12 bats likely would have continued beyond our predefined endpoint. Dex not only attenuates inflammatory responses but has been suggested to mimic naturally occurring stress (i.e., production of cortisol and other GCs by the adrenal cortex)[32,50]. In wild bat species, increased GC production has been linked to seasonality, breeding/birthing cycles and/or habitat disturbances along with changes in immune cell populations[51,52]. Juvenile ERBs are the most prevalently MARV-infected within wild colonies, with seasonal MARV pulses and concomitant recorded MVD outbreaks correlating to bat breeding/birthing cycles, when more naïve juveniles are available to perpetuate virus circulation[48]. In these tightly-packed, highly-populated colonies usually confined to caves or mines, it is likely that many juvenile ERBs, following maternal weaning, in times of food scarcity and/or possibly primed by coinfections with native heterologous pathogens (e.g., Sosuga or Kasokero viruses[37]), experience stressor conditions as they learn to self-forage. Indeed, nutritional stress has previously been associated with higher Hendra virus seroprevalence and shedding in *Pteropus* bat populations[53,54]. Together, our findings suggest a compelling scenario in which environmental stress could weaken aspects of wild ERB immunity (e.g., inflammatory responses akin to Dex), fomenting recurrent MARV super-shedding events that enhance juvenile bat transmission and ultimately risk of spillover to unprotected hosts, including humans[3].

## Methods

### Biosafety and animal care and use

Experiments with animals and viable Marburg virus (MARV) complies with all relevant regulations, with study protocols (#2977, #3090) overseen and approved by CDC's Institutional Animal Care & Use Committee (IACUC), Animal Care and Use Program Office (ACUPO), Comparative Medicine Branch (CMB) and Institutional Biosecurity Board (IBB), using guidelines established by the Association for the Assessment and Accreditation of Laboratory Animal Care,

International (AAALAC), the Animal Welfare Act and Regulations, and The Guide for the Care & Use of Laboratory Animals[9,11,12,48]. All procedures with live bats and virus were performed in biosafety level 4 (BSL4) high containment by highly trained laboratorians. Animal husbandry was carried out daily by trained animal caretakers. Cages housing bats were contained within bio-flow isolator units with HEPA-filtered intake and exhaust manifolds (Duo-Flow Mobile Units, Lab Products, Inc.). Animal housing rooms were climate controlled with a 12h day/night light cycle. Fresh fruit was provided daily and water was provided ad libitum. Leather bite gloves were used for all animal handling. Proper infection control practices were in place to prevent cross-contamination between bat cohorts.

## MARV infection and immunomodulation of ERBs

33 male ($n = 20$) and female ($n = 13$) juvenile (8–10 month old), captive-bred Egyptian rousette bats (ERBs, *Rousettus aegyptiacus*), originating from an established, MARV-naïve, multi-generational colony, were used in this study; juveniles were used as they are the most prevalently MARV-infected ERBs in the wild[9,11,12,48]. The study included multiple bat cohorts: negative (naïve uninfected) control bats (Neg, $n = 5$), dexamethasone (Dex)-treated uninfected bats (Dex, $n = 5$), MARV inoculated bats euthanized at 6DPI (M6, $n = 5$) or at 12DPI (M12, $n = 4$), simultaneously Dex-treated and MARV-inoculated bats euthanized at 6DPI (DM6, $n = 5$) or at 12DPI (DM12, $n = 5$), and finally MARV-inoculated bats that were Dex-treated at 6DPI and euthanized at 12DPI (D6M12, $n = 4$). Animals were assigned to cohorts randomly, with an effort to distribute males and females as equally as possible among cohorts, although sex was not considered in our study design or analysis given a lack of evidence for any sex-based differences in MARV infection or related immune responses in this species experimentally or in the wild[7,9,11,12,48,49,55]. Bats were group-housed and acclimated for at least 5 days in the BSL4 prior to manipulation. For all MARV inoculations of ERBs, the well-characterized bat isolate MARV-371bat (Gen-Bank: FJ750958.1), originally isolated from a wild-caught ERB from the Kitaka mine in Uganda and subsequently passaged twice on Vero E6 cells[7,48], was diluted in 250 μl of sterile medium and administered subcutaneously at $10^4$ TCID$_{50}$ (50% tissue culture infectious dose) under isoflurane inhalational anesthesia; uninfected bats were mock-inoculated with sterile medium only. For all chemical immunomodulation (IM) of ERBs, we administered Dex (2 mg/ml, VetOne) to ERBs intraperitoneally under anesthesia. ERBs were either mock-IM (administered saline only) or administered stepdown dosages of Dex every other day on day (d)0 (25 mg/kg), d2 (12.5 mg/kg) and d4 (6.25 mg/kg). Dex treatment occurred alone (Dex cohort), initiated simultaneously with MARV inoculation (DM6 and DM12 cohorts) or initiated 6DPI (D6M12). Uninfected IM (Dex) and mock-IM (Neg) bats were euthanized 8 days after the first Dex/mock dose, while bats inoculated with MARV (with and without Dex) were euthanized at either 6DPI or 12DPI. Stepdown alternate day dosages and regimen were analogous to and adapted from those described in previous successful IM studies in rodents[28,29,31], which we modified to account for minimal bat body weight and by administering intermediate drug concentrations in order to balance having the best possible efficacy in ERBs with elimination of maintenance dosing (due to our shorter study duration, to minimize the number of animal manipulations requiring anesthesia and IP injections, and to reduce the potential risk of environmental [Dex-related] secondary infection in IM bats).

## Bat sampling, complete blood counts, necropsy and virus quantitation

Routine and endpoint bat procedures were performed throughout the study[9,11,16,48]. Briefly, wing blood sampling to measure viral RNA loads was conducted daily post-inoculation, with pre-bleeds obtained prior to the study to establish baseline values. Rectal probing was performed daily to obtain temperatures. For bat cohorts with endpoints at 12DPI,

oral and rectal swabs were routinely collected to monitor for virus shedding, with rectal thermometer probe covers doubling as swabs; pre-swabbing was also performed prior to inoculation. All blood and swab samples were collected in MagMAX lysis buffer (ThermoFisher Scientific). Complete blood counts (CBCs) were performed on an Abaxis VetScan HM5 (Zoetis) with software ver. 2.4 to monitor IM efficacy, using blood collected in microvette capillary tubes containing EDTA (Sarstedt). Weights were obtained regularly to monitor for any clinical signs of IM toxicity or IM-induced MARV-mediated disease. Following euthanasia by isoflurane overdose and cardiac exsanguination, bats were immediately dissected and ~100 mg pieces of various tissues were placed in MagMAX lysis buffer. Total RNA was extracted by the MagMAX-96 Total RNA Isolation kit (blood/swabs) or MagMAX-96 Pathogen DNA/RNA kit (tissues), run on a MagMAX Express-96 processor, and used for quantitative reverse transcription polymerase chain reaction (qRT-PCR) on an Applied Biosystems 7500 Real-Time PCR system with software ver. 2.0.6 (Thermo) with MARV *VP40*-specific primers/probe, positive and negative control samples and *18S* reference gene primers/probe. For daily blood viral loads, quantitation was reported as the geometric mean cycle threshold (Ct) value. For all other viral quantitation, a standard curve was generated using serial dilutions of a known infectious quantity of MARV-371bat and used to extrapolate TCID$_{50}$ equivalent amounts in the samples. To determine blood chemistries, heparinized plasma from terminal cardiac blood was separated and 105 μl per sample was run on an Abaxis Piccolo Xpress Chemistry Analyzer with software ver. 3.1.37 (Zoetis) using General Chemistry 13 Panel discs.

## Flow cytometry

To isolate and prepare total splenocytes and axillary lymph node (ALN) cells, tissues were placed in gentleMACS C tubes containing Spleen Dissociation Kit (mouse) enzyme mix (Miltenyi Biotec) and run on a gentleMACS Octo Dissociator with Heaters (program 37C_m_SDK_1). RPMI supplemented with 10% FBS and Benzonase was added to cell suspensions and passed through 70 μm strainers into conical tubes. Pellets were resuspended in ACK Lysing Buffer (Thermo) and incubated at room temperature (RT), washed and resuspended in RPMI/Benzonase, and finally single-cell suspensions were enumerated using an ORFLO MOXI Z Mini cell counter. ~2 × 10$^6$ splenocytes and ALN cells were stained with LIVE/DEAD Fixable Aqua viability dye (Invitrogen, 1:1000 dilution), and then splenocytes were subsequently stained with the following fluorescently-conjugated antibodies diluted in PBS + 2% rat serum: anti-CD11b-APC/Fire 750 (BioLegend, clone ICRF44, 0.25 μg/sample), ERB-specific anti-CD14-PE and anti-CD19-APC (previously custom generated and validated in partnership with the U.S. Army Medical Research Institute of Infectious Diseases, with rat immunizations and hybridoma development having been performed by Aldevron, LLC[9,23], 1 μg/sample), and anti-MHCII-BV786 (BD Biosciences, clone 2G9, 1 μg/sample). Anti-CD14 and anti-CD19 fluorophores were conjugated using the respective Lightning-Link conjugation kits (Novus Biologicals). After washing, cells were fixed in CytoFix (BD Biosciences). Following an overnight incubation, the fixative was replaced with PBS + 2% rat serum and flow cytometry was performed using the Stratedigm S1400EXi cytometer platform with CellCapTure software ver. 4.1 (Stratedigm, Inc). Gating strategy is shown in Supplementary Fig. 2. Briefly, using FlowJo 10 software (TreeStar), data analyzed blindly were gated first by removing debris, then selecting for singlets. Cell viability (splenocytes and ALN cells) was then assessed by gating to exclude positive live/dead-stained cells, compared to total cells with forward scatter (FSC) and side scatter (SSC) characteristics of total leukocytes. Percent of viable cells was gated by comparing live cells verses total leukocytes and was used to calculate the total viable cells per tissue using the absolute cell counts from the dissociation of splenocytes/LN cells. Splenic CD11b$^+$ and CD14$^+$ cells were then determined as a percentage of total viable cells, after gating on live

cells and total leukocytes. Splenic B cells (CD19$^+$) were determined by gating on live cells, then cells with FSC/SSC characteristics typical of lymphocytes, and finally the CD19$^+$ proportion. Splenic T cells (CD19$^-$) were estimated by calculating the remaining cells in the lymphocyte gate after gating out the CD19$^+$ population, which likely also include other lymphocyte subtypes (NK and NKT cells).

## Histology

Bat carcasses were submerged in 10% buffered formalin in individual containers for a minimum of seven days, including one intermediate exchange with fresh formalin, after which carcasses were removed from the high-containment laboratory. Within a week, the formalin was replaced with 70% ethanol. Representative sections from the following tissues were obtained for histopathology: liver (four non-contiguous sections), spleen (two cross sections at the hilus), lymph nodes (axillary, mesenteric, tracheobronchial and submandibular), lung, heart, thymus, stomach, gall bladder, urinary bladder, intestines, colon, pancreas, kidney, adrenal gland, reproductive organs, salivary gland, brain, eye, tongue and skin from the inoculation site. Tissue sections were routinely processed, embedded in paraffin, sectioned at 4–5 μm, mounted on glass slides, stained with hematoxylin and eosin (H&E), and initially reviewed blindly using randomly numbered samples by a veterinary pathologist (S.G.M.K.).

## Immunohistochemistry (IHC)

IHC was performed on formalin-fixed paraffin embedded (FFPE) livers, spleens and ALNs at the University of Georgia Histology Laboratory using the following primary cell marker antibodies: mouse monoclonal anti-CD3 (Dako/Agilent Technologies, clone F7.2.38, 1:1000 dilution), mouse monoclonal anti-CD79a (Biocare Medical, clone HM47/A9, 1:50 dilution), rabbit polyclonal anti-ionized calcium-binding adapter molecule 1 (Iba1, Wako, Cat #019–19741, 1:8000 dilution) and rabbit monoclonal anti-Ki-67 (Cell Marque Corporation, clone SP6, 1:50 dilution). 3,3′-Diaminobenzidine (DAB, Biocare) was used as the chromogen (brown stain) for all IHC protocols and sections were counterstained with hematoxylin. Positive control slides were included in each IHC run using non-bat tissues known to contain cells immunopositive for each marker. Appropriate immunostaining in bat tissues was confirmed by visualization of immunopositive cells with the expected morphology and location for each immune cell marker (i.e., positive internal controls). CD3, CD79a, Iba1 and Ki-67 IHC slides were scanned with an Aperio ScanScope XT (Aperio Technologies, Inc.) and Aperio AT2 Console ver. 102.0.7.5 (Leica Biosystems) at ×20 magnification. IHC evaluation, morphometry and quantitation were initially performed blindly using randomly numbered samples. Tissues for each bat were digitally traced using the manual annotation tool in Aperio ImageScope software ver. 12.4 to set the region(s) of interest (ROIs). Background staining and artifacts were eliminated from the ROIs. CD3, CD79a and Iba1-stained slides were loaded into ImageScope for quantitation. The total absolute number of positive pixels (per cross-section to account for mild to moderate changes in tissue size due to expansion of splenic immune cell lineages following MARV-related antigenic stimulation or to account for splenic hemorrhage, or per section to reflect overall Dex-induced reductions in spleen/ALN size) or percentage of positive pixels (normalized data to account for variation in tissue cross-sectional areas in liver and/or spleen) was quantified using the Positive Pixel Count (PPC) algorithm ver. 9.1 with the default hue (0.1) and width (0.5) parameters. The color saturation threshold (CST) was optimized for each antibody to minimize the detection of background staining (CST range = 0.04–0.09). Percent decreased IHC staining in digital tissue images were calculated by dividing the average absolute pixel count from one bat cohort by the average absolute pixel count from another cohort, subtracting that value from 1 and multiplying by 100.

## In situ hybridization (ISH)

ISH was performed on FFPE ERB tissues either at Advanced Cell Diagnostics (ACD, automated assay kits) or in-house (manual assay kits) using the manufacturer's protocol for the RNAscope duplex assay[56]. Custom antisense RNA probes were designed to target MARV viral protein gene *VP40* and nucleocapsid protein gene *NP* (GenBank: FJ750958) and predicted ERB mRNA sequences for tumor necrosis factor (*TNF*; XM_016121699), interleukin-6 (*IL6*; XM_016159525), *IL10* (XM_016129276), *CD3e* (XM_016155454), *CD79a* (XM_016129923) and *CD14* (XM_016166396); see Supplementary Table 1 for additional probe information. Duplex ISH pairing MARV with each cytokine and immune cell marker listed above were carried out on liver, spleen and ALN. ISH and quantitative analyses were subjected to a rigorous quality control process. A positive control probe targeting an ERB medium-expressing housekeeping gene was applied to tissues from at least one bat per cohort to confirm RNA integrity and to confirm consistency of expression in the liver and spleen across individuals and cohorts (peptidylprolyl isomerase B, *PPIB*, XM_016141088.1). A probe targeting the bacterial *dap*B gene was used as negative control for each ISH run to exclude non-specific staining of the probes. Paired MARV and cytokine assays were performed on all tissue samples, regardless of infection or treatment status, to further demonstrate a lack of non-specific probe binding. Standard and extended pretreatment conditions were used for spleen/ALN and liver, respectively, as per the manufacturer. MARV hybridization signal was detected using green chromogen, while hybridization signal for each cytokine and immune cell marker was visualized using red chromogen, and sections were counterstained with hematoxylin. Slides were scanned with an Aperio ScanScope XT at ×40 magnification, loaded into QuPath ver. 0.4.1 and manually annotated to set ROIs and eliminate background staining[57]. Whenever possible, and for the majority of samples analyzed, ROIs included multiple sections of liver (at least 10 mm$^2$ liver tissue) and two sections of spleen (at least 3 mm$^2$ splenic tissue) for the quantitative cytokine analyses. Cytokine hybridization signal in the liver and spleen was quantified using the Create Threshold tool (threshold range = 0.15–0.5), and appropriate staining was verified for each sample. For each ROI, cytokine expression data were expressed as the percentage of positive pixels to normalize the data for comparison across individuals and cohorts (to account for variation in the tissue area analyzed). To better visualize virus at low magnification and in tissues potentially important in viral shedding, single-plex RNAscope ISH assays (ACD, manual assay kit) were performed using the MARV *VP40*/*NP* probe on the liver, spleen, ALN and non-primary target tissues, as available (brain, eye, tongue, kidney, urinary bladder, adrenal gland, salivary gland, lung, trachea, esophagus, stomach, intestines, colon, reproductive organs, pancreas and skin) of at least one representative bat from Neg and all MARV-infected cohorts using the RNAscope 2.5 HD Assay−Red according to the manufacturer's protocol. ERB *PPIB* and *dap*B probes were used as positive and negative controls, respectively. For all ISH assays, bat tissues were assigned a semi-quantitative score: 0 = MARV not detected, 1 = minimally MARV$^+$ (1–10% of tissue), 2 = occasionally MARV$^+$ (11–25% of tissue), 3 = frequently MARV$^+$ (25–75% of tissue), and 4 = extensively MARV$^+$ (>75% of tissue). Full ISH scoring of individual bats and tissues can be found in Supplementary Table 2.

## Codetection of mononuclear phagocytes and MARV

MARV RNA and cells of the mononuclear phagocyte (MNP) lineage were codetected in a DM6 bat (*n* = 1, all systemic tissues) by combination ISH-IHC, using the MARV *VP40*/*NP* RNAscope probe and Iba1 antibody, respectively. First, RNAscope was performed on each slide according to the manufacturer's protocol (ACD, manual assay kit) as described above. Next, IHC was initiated by applying a universal blocking reagent (Background Punisher, Biocare) and then anti-Iba1 was used as described above (diluted in Co-Detection antibody Diluent

[ACD]). Slides were incubated with MACH 3 Horseradish Peroxidase (HRP) Polymer (Biocare) followed by a drop of HRP Chromogen (Biocare). Finally, slides were counterstained with hematoxylin. Appropriate immunolabeling was verified using tissue macrophages as internal controls.

### Statistics and reproducibility

To graph MARV RNA loads in tissues, splenocytes and oral/rectal mucosa (shedding) between the different bat cohorts, we plotted the $log_{10}$ geometric mean and geometric standard deviation (SD). To graph the cytokine ISH quantifications, we plotted the $log_{10}$ mean % positive pixel values and standard error of the mean (SEM). All other graphs are plotted as the mean and SEM. To assess statistically significant differences between bat cohorts, we applied two-tailed Student's $t$ tests for each grouping. GraphPad Prism ver. 9.5.0, Microsoft Excel 365 ver. 2208 and Microsoft PowerPoint 365 ver. 2308 were used for data visualization and statistical analysis. Data reported herein was highly reproducible between bats with few notable exceptions. H&E and IHC-based (Iba1, CD3e, CD79a and Ki-67) stains of the liver, spleen and ALN were similar amongst all bats included within each study cohort (except DM12 bats as specified below and Iba1-stained ALN of Dex bats [$n = 4$]); IHC was not performed on D6M12 bats as their H&E-stained livers and spleens appeared similar to M12 bats. Three DM12 bats had reproducible microscopic findings consisting of severe hepatic necrosis ($n = 2$) and/or hepatic parenchymal loss ($n = 1$), a ductular reaction ($n = 3$), and mononuclear cell inflammation ($n = 3$); these were designated high-replicators (high-rep). The two remaining DM12 bats had more limited microscopic changes in the liver, similar to M12 bats, and were designated low-replicators (low-rep). IHC tissue staining was similar among high-rep DM12 bats and among low-rep DM12 bats. Meanwhile, duplex ISH assays were performed to colocalize immune cell types (*CD14*, *CD3e* and *CD79a*) with MARV RNA in the liver, spleen and ALN from one bat in each of the Neg, Dex, M6 and DM6 cohorts. Subsequently, duplex ISH was performed to colocalize cytokine (*TNF*, *IL10* and *IL6*) mRNA with MARV RNA in all available bat livers and spleens (Neg $n = 4$, Dex $n = 5$, M6 $n = 3$ [liver] and $n = 4$ [spleen], M12 $n = 4$, DM6 $n = 2$ [liver] and $n = 3$ [spleen], high-rep DM12 $n = 3$, low-rep DM12 $n = 2$). MARV RNA was not detected in any Neg or Dex bat tissue ($n = 4$ and 5 bats/cohort, respectively). ISH results were similar among bats within each MARV-infected cohort except high-rep versus low-rep DM12 bats; cytokine expression and MARV RNA of low-rep bats was similar to M12 bats, whereas high-rep bats showed consistently higher cytokine expression and MARV RNA. Finally, single-plex ISH was additionally performed in the liver and non-primary tissues as described above (M6 $n = 1$, M12 $n = 1$, DM6 $n = 2$, high-rep DM12 $n = 3$), with similar findings among replicate bats within DM6 and DM12 cohorts for all tissues tested. Individual semi-quantitative MARV ISH scores ($n = 111$ ISH slides) are provided by tissue in Supplementary Table 2.

### Reporting summary

Further information on research design is available in the Nature Portfolio Reporting Summary linked to this article.

## Data availability

Data supporting this work are available within this manuscript and Supplementary Information. Source data are provided with this paper.

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

## Acknowledgements

The authors wish to thank members of CDC's Comparative Medical Branch for their assistance with bat husbandry in the BSL4 and Sarah Genzer of VSPB/CMB for assistance in complete blood count assay optimization; members of the University of Georgia (UGA) and Emory National Primate Research Center Histology Laboratories for tissue embedding and slide preparation; and Jian Zhang of UGA, Elizabeth Lee, Jana Ritter and Julu Bhatnagar of CDC's Infectious Disease Pathology Branch, and ACD for assistance with the ISH assays. Funding for this study was provided by CDC and RKI core funding. Opinions, interpretations, conclusions and recommendations are those of the authors and are not necessarily endorsed by the CDC or RKI.

## Author contributions

J.C.G., J.B.P., S.G.M.K. and J.S.T. conceptualized and designed the study. J.C.G., S.G.M.K. and J.B.P. prepared the manuscript, while all three coauthors and J.S.T. provided critical review and revisions of the drafts and approved the decision to submit for publication. J.C.G., J.B.P., B.R.A., A.J.S., J.G., J.R.S., J.R.H. and T.K.S. performed BSL4-based experimentation including immunomodulation, inoculation, sampling and/or necropsy; downstream specimen processing; and/or staining for flow cytometry. J.C.G. and J.B.P. performed flow cytometry sample processing; D.M.W. further analyzed flow cytometry data. T.K.S. performed RNA extractions and PCR-based processing; J.C.G. and J.B.P. further analyzed PCR-based results. S.G.M.K. performed all histological analyses, including IHC and ISH. J.C.G., S.G.M.K., J.B.P. and J.S.T. interpreted the results. All authors have read and agreed on the content of the manuscript.

## Competing interests

The authors declare no competing interests.
