## [Peer Review File · Nature Communications]

Coordinated inflammatory responses dictate Marburg virus control by reservoir batsEditorial Note: This manuscript has been previously reviewed at another journal that is not operating a transparent peer review scheme. This document only contains reviewer comments and rebuttal letters for versions considered at *Nature Communications*.

REVIEWER COMMENTS

Reviewer #1 (Remarks to the Author):

The authors have appropriately addressed the outstanding questions

Reviewer #2 (Remarks to the Author):

In this revised manuscript by Guito and colleagues, they have improved the presentation and addressed all the minor suggestions raised during the review of the original submission. The authors were not able to address the major shortcoming of the study in investigating the detailed mechanism of the central role played by an immune response elicited by ERBs in maintaining asymptomatic infection during experimental MARV infection as "spatial transcriptomics panels currently are not available for ERBs, and the technology remains cost-prohibitive based on our lab's budgetary constraints."

Reviewer #3 (Remarks to the Author):

This is an interesting study that will enable the bat research community to develop additional in vivo models and tools to study virus-host interactions. However, the study needs additional revisions to address the issue of lack of scientific rigour.

1. The authors state that similar cell type markers have been used to define immune cell subsets in other bats; however, bats are over 1460 different species and it is hard to presume that all bat species will likely have similar immune cell type markers and/or composition. However, the authors do mention that these cell type markers evolved prior to the divergence of bat species. This needs to be validated with rigorous citations to demonstrate scientific rigour in this manuscript.

2. Differences in cytokine levels by qRT-PCR need to be demonstrated after normalization to a housekeeping gene. This is necessary for scientific rigour of the data. Multiple housekeeping genes have been used in other bat studies. The data need to be normalized by using transcript levels of housekeeping gene/s that do not fluctuate with Dex treatment and/or infection.

3. The authors state that, 'Perhaps the MARV-infected cells are secreting another pro-inflammatory cytokine that we did not measure which is causing a subset of the cells around the infected cells to secrete TNF and/or IL10. Or perhaps they were infected and subsequently cleared the viral RNA, but are still expressing low levels of cytokine mRNA'. This needs to be discussed in the discussion section of the manuscript.

Reviewer #3 (Remarks to the Author):

This is an interesting study that will enable the bat research community to develop additional in vivo models and tools to study virus-host interactions. However, the study needs additional revisions to address the issue of lack of scientific rigour.

1. The authors state that similar cell type markers have been used to define immune cell subsets in other bats; however, bats are over 1460 different species and it is hard to presume that all bat species will likely have similar immune cell type markers and/or composition. However, the authors do mention that these cell type markers evolved prior to the divergence of bat species. This needs to be validated with rigorous citations to demonstrate scientific rigour in this manuscript.

We thank the Reviewer in pointing out the potential for further clarification here. We are a little unsure of whether the Reviewer is 1) suggesting that the immune system is generally comprised of the same cell types, but we cannot be sure whether the markers we choose correctly identify the cell types we assume they do, or 2) whether the Reviewer is suggesting that there are fundamental differences between the immune system of bats (ERBs specifically) and other mammals that have been characterized (a T cell in a bat/ERB is fundamentally different from that of a mouse/human, for example). We assume that the Reviewer is suggesting the former.

In some cases, where a marker might be used that only suggest a cell type but is not inherent to the function of that cell type, we would generally agree with the Reviewer's concerns. In our case, we used the annotated ERB transcriptome to design probes that target gene or protein markers that are distinctive and integral to the functions of the cells type in question. For T cells, for example, we designed probes against CD3e, which is a major functional unit of the T cell receptor complex itself, so by definition, cells that express this 'marker' would be a T cell. CD79A and B make up the B cell receptor - so again, this would be expressed in a B cell. Studies exemplifying evolution of the immune system typically use these genes as “anchor genes” to identify these cell types and to look at how variable regions of immune-related genes and regulatory factors differ and have evolved between vertebrates:

<https://www.sciencedirect.com/science/article/pii/S2090123223000693>.

CD14 is a co-receptor with TLR4, so functionally detects LPS and pathogens and its expression would indicate a phagocytic innate immune cell. CD14 is highly conserved, however, in its regulation and differences in downstream functions are what is sometimes variable between species.

<https://www.ncbi.nlm.nih.gov/pmc/articles/PMC6628885/>

<https://www.ncbi.nlm.nih.gov/pmc/articles/PMC6628885/>

Iba1 is very highly conserved across mammals and has been well studied for its expression patterns in specific cell types: <https://pubmed.ncbi.nlm.nih.gov/11943136/>. It is a marker widely used by both human and veterinary pathologists to identify histiocytic lineage cells not only in mammalian tissues but also in a variety of non-mammalian vertebrates. It is common and accepted practice for veterinary pathologists to apply this antibody and other antibodies directed at highly conserved targets to tissues from a variety of animal species both in diagnostic and research settings. Veterinary pathologists receive extensive training in the comparative anatomy, histology, physiology and immunology of a wide variety of animal species, including both vertebrates and invertebrates, and thus, are well positioned to validate appropriate staining (i.e., expected cell types in tissues) of reliable and highly conserved cell markers in novel animal models. In fact, this is a major function of many veterinary pathologists employed by research and/or molecular pathology laboratories.

Furthermore, the staining patterns, spatial position and morphological characterization of these cells aligns precisely with what one would expect. For example, the morphology and tissue distribution of T, B and macrophages in the spleen, and the detection of KCs in the liver, which are distinct and easily identified by a pathologist (Dr. Kirejczyk), are consistent with these cell types across vertebrates.

For the ERB-specific antibodies, these have been validated in multiple past publications that are already cited in this manuscript.

The cytokines themselves, are some of the most conserved and of ancient origin:

<https://www.ncbi.nlm.nih.gov/pmc/articles/PMC4863567/>

Thus, we are highly confident that the markers chosen are correctly identifying their intended specific cell types in ERBs, which along with cytokines, are highly conserved in vertebrates. The question that remains is how the immune system is regulated in bats/ERBs and what is unique in response to MARV infection, for which much work is to be done. A T/B/Mac would have generally similar functions, but of course differences surely lie in their regulation and nuanced activities during innate and adaptive immune responses. These are primarily transcriptional regulation, not overt differences in the function of a specific cell type (see Figure 1 of <https://immunityageing.biomedcentral.com/articles/10.1186/s12979-021-00232-1>).

This complex interplay between immune cells and pathogens must be unraveled before a complete understanding of how natural reservoirs live with the viruses they harbor. Here we demonstrate that T cells are recruited to sites on MARV infection and a localized and regulated inflammatory response ensues. Driving the response ectopically towards an anti-inflammatory response via Dex and ablating this T cell response leads to unchecked MARV replication and tissue pathology, and disease in some ERBs.

To better clarify our points above, we have added additional context to the manuscript, which can be found on lines 90, 102-106, 276-277 and 414-417, with accompanying additional citations (Jiao 2024, Riera Romo 2016 and Quistad 2016: <https://www.nature.com/articles/cddiscovery201658>).

2. Differences in cytokine levels by qRT-PCR need to be demonstrated after normalization to a housekeeping gene. This is necessary for scientific rigour of the data. Multiple housekeeping genes have been used in other bat studies. The data need to be normalized by using transcript levels of housekeeping gene/s that do not fluctuate with Dex treatment and/or infection.

We appreciate the concern raised and would like to further discuss our rationale for the *in situ* hybridization (ISH)-based approach to investigating and quantifying cytokine mRNA expression (as opposed to using qRT-PCR for this purpose, which we instead only performed to analyze viral loads), and to provide additional details of the methods undertaken, which we believe bolster the scientific rigor of our approach.

We chose a quantitative molecular pathology (ISH)-based approach to rigorously explore cytokine expression in non-infected and MARV-infected ERBs for two reasons.

First, there is clear microscopic evidence of an immune response in the livers of MARV-infected ERBs, yet we previously were unable to detect differences in many cytokine mRNA levels using the NanoString custom ERB immune-related gene panel and nCounter analysis (Guito, et al. 2021, *Curr Biol*). This is surely, as we know now, because relatively few cells in these tissues regulate cytokine mRNAs to a large extent and this is surrounded by large areas of normal tissue. Thus, the signal-to-noise ratio is not in favor for detecting these genes as differentially expressed, and this should prove the same for qRT-PCR. The low-level positive ISH signal observed on the RNAscope ISH-stained slides from Neg bat, Dex bat and M6/M12 bat livers and spleens corroborate the NanoString data that cytokine expression is very low in both naive and MARV-infected bats.

Second, since we previously did not see many immune-related changes at the whole tissue level by examining mRNAs, we wanted to more precisely investigate tissues histologically (and therefore spatially). Because a large-scale spatial transcriptomics approach was not feasible, we chose to develop specific markers of important cell types and pro- and anti-inflammatory cytokines to assess ERB transcriptional responses spatially and specifically at the sites of MARV replication – something that qRT-PCR cannot achieve.

Having laid out the rationale for why we used an ISH-based approach to spatially explore ERB cytokine expression in the context of MARV infection, we respectfully disagree with the Reviewer's view that this expression data is insufficient and thus leads to biased assumptions for several reasons. We have included many of the below points as additional clarifying details within the Methods section (lines 578-597).

First, not only are the RNAscope probes extremely sensitive and specific for the chosen cytokine mRNA and viral targets, but the positive cytokine ISH signal is present within the appropriate and expected context within the liver and spleen. In the liver, the positive *TNF* and *IL10* ISH signals are contained within cells morphologically consistent with mononuclear phagocytes (presumably Kupffer cells in the sinusoids and monocyte-derived macrophages recruited to foci of MARV infection).

Second, strict, standardized practices were adhered to in order to minimize potential bias and enhance scientific rigor throughout the data collection process, from sample selection and tissue processing, to running the ISH assays, to the use of appropriate control ISH probes, slide scanning and annotating, and finally, to quantitating the mRNA signal and comparing cytokine expression levels between cohorts. qRT-PCR-based expression data collected from a piece of tissue lack the crucial spatial context that ISH-based analyses afford. For this study, we selected four random pieces of liver and two pieces of spleen and quantified the ISH signal from at least 10 mm² liver and 3 mm² spleen for each tissue type whenever possible to ensure representative samples were obtained since MARV replication foci are so sparse in immunocompetent ERBs. The quantitation algorithms were not blindly applied to the whole slide images in the analysis software; rather, one of the authors (Dr. Kirejczyk, a board-certified veterinary pathologist) verified appropriate staining for each of the probes throughout all the tissues (a very labor-intensive process that is considered the standard in the pathology community).

Third, when performing RNAscope ISH, negative and positive control assays were run using *DAPB* (negative control; a gene from *Bacillus subtilis* that is not contained within the host tissues) and *PPIB* (positive control; peptidyl-prolyl isomerase B, a medium-expressing housekeeping gene) to rule out non-specific staining and to verify the integrity of the RNA in the tissue samples, respectively, as included on lines 580-585 of Methods. Our ISH-stained slides were subjected to a rigorous quality control process, in which *PPIB* staining was shown to be consistent across cohorts, and it did not fluctuate with dexamethasone treatment or MARV infection. Qualitative evaluation of the *PPIB* is standard practice per the manufacturer's recommendation.

Fourth, we performed duplex ISH assays (paired MARV probe with each cytokine) on all samples, regardless of infection or treatment status, to demonstrate a lack of non-specific probe binding with the MARV probe in uninfected bats. These added controls further support the notion that, given the appropriate spatial context, the co-detection of cytokine mRNA ISH signal and MARV RNA ISH signal in MARV-infected bat tissues is an accurate representation of what is occurring locally in target ERB tissues following MARV infection and not a biased assumption.

Finally, ISH has been shown to be both highly sensitive and specific for the detection of mRNA targets in formalin-fixed paraffin-embedded tissues (Wang, et al., 2012). Computer software-based detection of individual RNAscope "puncta" in tissue sections have been shown to correlate well with expression of the corresponding genes in tissue culture, as detected by qRT-PCR (Wang et al., 2012); using the fluorescent detection kits, each RNAscope puncta or dot is reported to represent a single mRNA copy. Because we chose the chromogenic assay for our study, individual puncta are not always present in cells with positive signal; for this reason, we presented the cytokine expression data as the percentage of the tissue area staining positively for each cytokine probe. While we cannot translate the percentage of positive pixel data to individual mRNA copy numbers per cell, presenting the cytokine expression data as the percentage of positive pixels serves to normalize the data across individuals to account for slight variations in the tissue areas analyzed. We presented non-normalized IHC-based immune cell quantitative data when we reported the absolute positive pixel counts for the immune cell counts in the spleens, which dramatically changed size

following dexamethasone treatment (due to loss of immune cells/mass); in this case, it would not have made sense to normalize those data, and doing so would have led to a loss of biological context for the data.

We hope that Reviewer #3 will consider these points we have raised regarding the scientific rigor of the ISH-based cytokine analysis, and in our opinion, the lack of utility for a qRT-PCR-based cytokine analysis in the context of this study. Unlike qRT-PCR, RNAscope ISH-based analysis of cytokine expression provides the crucial advantage of both biological and spatial context in our samples.

Wang F, Flanagan J, Su N, Wang LC, Bui S, Nielson A, et al. RNAscope: a novel in situ RNA analysis platform for formalin-fixed, paraffin-embedded tissues. *J Mol Diagn.* 2012;14(1):22–29. doi: 10.1016/j.jmoldx.2011.08.002.

3. The authors state that, 'Perhaps the MARV-infected cells are secreting another pro-inflammatory cytokine that we did not measure which is causing a subset of the cells around the infected cells to secrete TNF and/or IL10. Or perhaps they were infected and subsequently cleared the viral RNA, but are still expressing low levels of cytokine mRNA'. This needs to be discussed in the discussion section of the manuscript.

We thank the Reviewer for the comment and agree that it would be beneficial to address this explicitly within the Discussion, which we've now included on lines 330-333.

REVIEWERS' COMMENTS

Reviewer #3 (Remarks to the Author)

Ok to accept this version.